# Treatment and Improvement of Healing after Surgical Intervention

**DOI:** 10.3390/healthcare11152213

**Published:** 2023-08-06

**Authors:** Andrea Bueno, Endika Nevado-Sanchez, Rocío Pardo-Hernández, Raquel de la Fuente-Anuncibay, Jerónimo J. González-Bernal

**Affiliations:** 1Health Center of Las Huelgas, 09001 Burgos, Spain; abuenof@saludcastillayleon.es; 2Reconstructive and Aesthetic Plastic Surgery Service, University Hospital of Burgos, 09006 Burgos, Spain; 3Propios Nevado Clinic, 09004 Burgos, Spain; 4Department of Health Sciences, University of Burgos, 09001 Burgos, Spain; raquelfa@ubu.es (R.d.l.F.-A.); jejavier@ubu.es (J.J.G.-B.)

**Keywords:** scars, surgery, treatments, healing techniques

## Abstract

The development of abnormal scars has a great impact on people’s well-being, and improving scarring outcomes after surgery is a field that currently lacks consensus. This review aims to identify newly researched approaches to improving the quality of surgical scars. A systematic search of PubMed, Scopus, Web of Science, and ScienceDirect was conducted between 13 May 2023 and 17 May 2023, in accordance with the recommendations of the PRISMA Statement. Study selection and analysis of methodological quality were performed in parts, independently and blindly, based on eligibility criteria. The 21 prospective, comparative, and randomized studies reviewed included 1057 subjects and studied approaches such as topical applications of creams with herbal extracts and silicone gels, growth factors, negative pressure dressings, oligonucleotides, intralesional injection of compounds such as botulinum toxin, skin closure techniques such as suturing and tissue adhesive, and laser treatments. There are recent research techniques that generate good results and are really promising to improve the results of surgical scars; however, the available evidence is extremely limited in some cases, and it is necessary to deepen its analysis to obtain reliable action protocols in each type of surgery.

## 1. Introduction

Scarring is the physiological response to skin damage. During the wound healing process, a phase of inflammation occurs, with an induction of the hemostatic cascade, leukocytes recruitment, and accumulation of fibrin [1,2]. This is followed by a stage of proliferation of fibroblasts that differentiate into myofibroblast precursors to generate local tissue adhesion, as well as keratinocyte proliferation and epithelization and the deposition of collagen fibers [1,3]. Finally, in the third stage, there is a remodeling of tissue until the final appearance is achieved. During this process, reepithelization ends, and the damaged dermis loses elastic fibers and hardens its collagen, generating greater firmness in the scar area [4,5,6].

Although scars are a part of the healing process, sometimes abnormalities such as hypertrophy or scar atrophy are generated, with the consequent increase in recovery time and physical and psychological impact for people [7]. To avoid these unwanted effects and improve clinical practice in the management of scars, scholars have tried to discover the biological mechanisms and factors that influence the recovery process [7,8,9]. Scar quality is influenced by factors such as age, infection, immune function, tissue oxygenation, nutrition, tobacco use, and particular circumstances such as the presence of diabetes, radiation, or chemotherapy [10].

For the treatment of abnormal scars and the improvement of the healing process, numerous investigations have been conducted on intervention or revision methods, and there are clinical management recommendations based on the available evidence. These include topical applications, intralesional medication (such as hyaluronic acid, corticosteroids, antimitotics), cryotherapy, laser applications, make-up camouflage, micro needling, radiofrequency, or oral agents [11,12,13,14,15].

Nevertheless, in wounds caused by surgical interventions, it is essential to know and meticulously plan the incisions, as well as the application of therapeutic measures that help reduce complications and improve the surgical outcome. These techniques include the surgical revision of the scars and subsequent treatments that promote proper healing. Based on previous scientific evidence, surgical revision of scars seems to show more reliable results, although there are countless advances in topical medications and treatments that could facilitate the therapeutic approach of surgical scars [16,17,18].

However, the application of treatments that improve the scarring outcome in surgical incisions is a field that still lacks consensus and is constantly updated by scientific progress and novel therapies that could bring benefits. This systematic review aims to identify approaches that are currently being investigated to improve the quality of surgical scars.

## 2. Materials and Methods

A systematic review of the scientific literature was conducted between 13 May and 17 May 2023, following the recommendations of the PRISMA Statement [19]. It began by formulating a research question (Table 1) in PIO format [20]. The electronic versions of the following databases were consulted: Pubmed, Web of Science, Scopus, and ScienceDirect.

To answer the question, different search strategies were used, adapted to the particularities of each database. Medical Subjects Headings (MeSH) and free text terms combined with Boolean operators AND/OR/NOT were included (Table 2).

We selected those original articles with a prospective longitudinal methodology, comparative or controlled, published in the last 5 years, carried out with humans, whose results evaluated the quality of scars caused by surgical interventions. Clinical case reports, scientific letters, bibliography reviews, those that analyzed other populations (animals) or etiologies, studies that do not answer the research question or were not related with the main objective of the review, and scientific reports of low quality were excluded.

Additionally, we carried out a manual reverse search, known as snowball-searching, in the bibliographic references of the studies included in the review, to identify possible relevant studies that had not previously been found through search engines.

The selection and evaluation of the methodological quality of the studies was carried out in pairs, blindly and independently. Discrepancies were solved by consensus, or failing that, through the participation of a third evaluator. The PEDro scale [21] was used to assess the methodological quality of the studies, considering a cut-off point of 8 points to accept the inclusion of each study in the review (Table 3). Each study was evaluated in terms of eligibility criteria specified, random allocation, concealed allocation, similarity of groups at the baseline, subjects blinding, therapists blinding, assessor blinding, less than 15% dropouts, intention to treat analysis, between-group statistical comparisons, and point measures and variability data.

A standardized data extraction form was designed in order to guarantee the homogeneity of the collected information, including the following aspects of the selected articles: principal investigator, year of publication, characteristics and sample size, implemented interventions, evaluation tools, and main results obtained, along with the results of their scientific quality evaluation.

## 3. Results

Of the 1738 studies initially identified, 21 were selected for systematic review after several phases of screening by automation, manual, and critical reading (Figure 1). The main characteristics of the selected studies are available in Table 4.

The studies comprised a total of 1057 subjects, with a range between 12 and 142. The female gender was predominant, with 702 women; however, some studies were designed only for one gender (studies within only females n = 4, and only male participants n = 1). Subjects of all ages were found as most of the reports analyzed an adult population, but four of them included children and teenager populations. All studies were comparative, with a control group (n = 12) or with other approaches and protocols (n = 9). A placebo was used in four of the studies. Follow-up time varied in selected studies from 1 to 12 months.

To assess the quality of the scars, standardized scales, clinical analyses, and measuring devices were used. The Patient Observer Scar Assessment Scale Score (POSAS) was used in 15 selected articles. It contains 2 domains with several subscales: Observer measure (vascularity, pigmentation, thickness, relief, pliability, and surface area), and Patient measure (pain, itchiness, color, stiffness, thickness, and relief). The Modified Stony Brook Scar Evaluation Scale (mSBSES) was used in 2 selected articles and includes width, height, color suture mark, overall appearance, and total score. The Vancouver Scar Scale (VSS) was used in 5 articles to assess the vascularity, pigmentation, pliability, and height. The Manchester Scar Scale (MSS) was used in 2 articles to assess color, matte/shiny, contour, distortion, and texture. The Hollander Wound Evaluation Score (HWES) was used in 1 trial, and includes step-off borders, contour irregularities, margin separation, edge inversion, excessive distortion, overall appearance, and total score. The Visual Analogue Scale (VAS) was used in 3 investigations to assess pain. Some of selected studies also assessed scar width (n = 1), patient satisfaction (n = 1), the skin elasticity coefficient (n = 1), and evaluations of elasticity, firmness, color, and composition with cutometer (n = 2), durometer (n = 1), mexameter (n = 1), and biopsies (n = 2), respectively.

According to these parameters, the characteristics that define the quality of the scars are their signs and symptoms. The selected studies had a large variability in the results (Table 4). Some reported improvements in total score [22,26,27,30,37,38,40,42,43], while others did so on specific subscales. Great variability was also observed in the type of surgery and in the interventions performed. The surgeries which subjects underwent were chest surgeries (n = 4), abdominoplasty (n = 2), surgical excisions (n = 3), tumor resections (n = 3), thyroidectomy (n = 3), epicanthoplasty (n = 1), trauma surgeries (n = 2), lumpectomy (n = 1), sternotomy (n = 1), and carpal tunnel relay (n = 1).

Among the studies analyzed, different suture materials were found. For the subcuticular closure, absorbable filaments were used, such as monocryl [28,29,32,36], polydioxanone [23], and vicryl [25,30,31,35,38,40]. Also, the skin sutures were made with nylon [24,32,35,39], Ethicon [23,28,29,30], and monocryl [36,40]. However, not all studies specified the materials used.

The implemented techniques include approaches which are carried out prior or at the moment of the original surgical procedure, such as botulinum toxin [24,35], tissue adhesive or sutures [25,31,33,39], laser treatments [30,32], and off-loading devices [41]; those which were implemented at the time of the intervention and are prolonged during the formation of the scar: off-loading devices [23], topical compounds [38,42], and negative pressure wounds [40] or those treatments that occur during the formation of the scar: botulinum toxin [22,27], topical compounds [26], off-loading devices [28], injected compounds [29], and laser treatments [36,37].

According to this, a great variability was observed at the time of implementation of the treatments, which are topical applications such as offloading devices, silicone, herbal extracts, pressure dressing or growth factors (n = 8); intralesional injections like botulinum toxin or oligonucleotides (n = 5); laser applications such as non-ablative fractional laser (NAFL), fractional carbon dioxide (FACL), pulse diode laser (PDL), or photobiomodulation (n = 4); and skin closure methods, like suturing techniques or tissue adhesive (n = 4).

Among the selected studies, research was conducted that analyzed different topical applications of creams, gel, and silicone sheets. Pangkanon et al. [34] studied the efficacy of a silicone gel with onion and aloe vera extracts, compared to silicone sheets to prevent the development of scar hypertrophy, without finding improvements in any of the variables analyzed, except in pliability (*p* = 0.009). Surakunprapha et al. [38] studied the effectiveness of a silicone gel with herbal extracts (which also includes onion extract) in sternotomy patients. At 6 months, the gel enriched with herbal extracts achieved improvements in vascularity, pigmentation, and overall opinion (*p* = 0.013; *p* = 0.000; *p* = 0.018) and the silicone group in vascularity and pigmentation (*p* = 0.046; *p* = 0.000), although intergroup comparisons are not known.

Other studies looked at topical application of compounds, such as that conducted by Dolynchuk et al. [26], who studied the topical use of 1,4-Diaminobutane (1,4DAB) compared to a topical control treatment (which does not contain 1,4DAB). A higher concentration of 1.4DAB was obtained in the biopsy analysis of the experimental group compared to the control, along with better results in firmness and scar quality (*p* < 0.05 each). On the other hand, Zoumalan et al. [42] found significant improvements compared to the control group with the cream enriched with growth factors in the measurements of the investigators, patients, and independent evaluators (*p* < 0.0001; *p* <0.001; *p* < 0.0001, respectively).

Furthermore, one of the selected studies investigated the efficacy of incisional negative pressure wound therapy on mastectomy scars. Timmermans et al. [40] found significant improvements at 3 months compared to the control group in total score, vascularity, and overall cosmesis, although the benefit was not maintained until 12 months by physician assessment. Patient assessment maintained improvements in color, pliability, thickness, and overall score at 12 months.

Other studies used adhesive devices on the skin. Chen et al. [23] found significant improvements in intragroup comparisons of the experimental group for width (*p* = 0.0025) and significant intergroup differences for scar irregularity (*p* = 0.0145) with a tissue adhesive zipper in children with surgical excision on the face. Ilori et al. [28] studied a heterogeneous group (excision tumors, open reduction fixation of fractures, osteotomies, arthroplasties) and found significant improvements in the type of scar, height, and width in the group treated with microporous tape over the scar compared to the control group (*p* < 0.0001 each), without having reported differences between the types of surgery. Zhang et al. [41] tested a discharge device to improve skin elasticity and promote healing in patients with a history of scarring hypertrophy with benign skin excisions. They found that there were no significant differences between the group that wore the device preoperatively and postoperatively, and those who wore it only after surgery; although there are significant improvements with respect to the control group (they did not use device) in terms of width, color, and overall opinion.

Another method found in the selected trials was the intralesional injection of compounds. Botulinum toxin was used by several authors with different results. Abedini et al. [22] recorded significant improvements at 3 and 6 months (*p* < 0.001) between their control (saline) and experimental (botulinum toxin) groups in mammoplasty and abdominoplasty patients; however, in intragroup comparisons, neither group improved significantly in width. In contrast, Chen et al. [24] found between-group differences in width and visibility (*p* < 0.01), getting better results with a high dose of botulinum toxin in patients with tumor resection. Huang et al. [27] found improvements after 1, 3, and 6 months (*p* = 0.034; *p* < 0.001; *p* < 0.001, respectively) over the control group (placebo) in their experimental group of botulinum toxin in epicanthoplasty patients. On the other hand, Phillips et al. [35] investigated the effect of botulinum toxin in thyroidectomy and found no significant differences between groups over time, although significant improvements were observed in POSAS (*p* = 0.012) and VSS (*p* = 0.021) between patients with a history of healing problems and those with a history of normal healing.

In studies with other injectable substances, Jensen et al. [29] performed well on mammoplasty scars after 24 weeks with their intervention using injectable anti-CTGF oligonucleotide compared to the control group in the evaluations of investigators (vascularity, pigmentation, thickness, relief, pliability, surface area, and overall opinion) and patients (color, stiffness, thickness, surface area, and overall opinion).

Another method found in the selected studies was the comparison of skin closure techniques. In one of the studies, Suwannaphisit et al. [39] compared the running subcuticular technique and the Donati technique of suture in patients with open carpal tunnel release, without finding significant differences between the two groups at 6 and 12 weeks.

In the same way, several selected studies analyzed the differences between suture and tissue adhesive. Musham et al. [33] found that tissue adhesive produced less pain than subcuticular suture (*p* < 0.01), but there was no significant difference in scar outcome at 1 and 3 months between groups in patients who underwent thyroidectomy. Kong et al. [31] also found no significant differences between groups in patients with bilateral hip arthroplasty, although from the point of view of patients, tissue adhesive was statistically better (*p* = 0.004).

However, Chung et al. [25] compared tissue adhesive application, suturing, and early non-ablative laser treatment (NAFL) in three experimental groups with thyroidectomy patients. Tissue adhesive was applied to the first group, which did not obtain any significantly better parameters in the intergroup comparisons. The second group received suturing and laser treatment, and significant improvements were observed for multiple subscales according to the researchers (thickness relief, pliability, surface area, and overall cosmesis); and the third group was treated with tissue adhesive and laser treatment, which also did not obtain significant improvements compared to the other protocols. However, patients reported greater satisfaction in the two laser-treated groups in all parameters except pigmentation.

On the other hand, the satisfaction of laser treatment was also observed by Ramos et al. [36] (PSAS *p* = 0.0047), who also found improvements in evaluator assessment (*p* = 0.0034) and VSS (*p* = 0.0065) with photobiomodulation after abdominoplasty. In the same way, Karmisholt et al. [30] found significant improvements in patients who underwent surgical excision with three sessions of NAFL in comparison with the control group (*p* < 0.001) and also noted that the best results with this technique were obtained in the chest area (*p* < 0.001). Safra et al. [37] also reported benefits in their experimental group of fractional ablative CO2 laser (FACL) with respect to the control group in lumpectomy patients (*p* < 0.001). Lin et al. [32] also studied the effects of applying FACL before suture or after the removal of sutures in patients with excision of skin cancer in extremities and found no significant differences in any of the scores, nor in biopsy analysis.

## 4. Discussion

From a global perspective, this systematic review aimed to explore current investigations of approaches to improve the quality of post-surgical scars. This review found that very promising approaches are currently being carried out to promote the correct healing process.

Among the methods found in the reviewed articles, topical applications of compounds, bandages, or off-loading devices stand out. Two of the articles applied silicone gel with herbal extracts with onion, compared to silicone gel [34,38], obtaining moderate improvements. A previous meta-analysis studied the effects of onion extract on healing, finding that onion gel increases adverse effects and has no benefit; however, onion extract in silicone gel generates improvements in healing and could be the optimal solution. Accordingly, the scientific evidence for silicone gel with herbal extracts for surgical scar enhancement is very scarce, and further research is needed to determine appropriate parameters such as doses and dosages that generate maximum benefit without harmful effects [44].

Other studies investigated the adhesion of devices to discharge mechanical stress from the surgical incision [23,28,41]. These mechanisms have been shown in animals to be associated with transcriptional downregulation of inflammatory pathways [45] and have obtained benefits in other similar research, helping to heal and improving their aesthetic appearance [46,47].

Similarly, a study using incisional negative pressure wound therapy was selected [40], achieving prolonged improvements in the opinion of the subjects, although according to the evaluation of the researchers the effect was not maintained until 12 months. According to a previous systematic review [48], this is a technique with very limited research and uncertain results, although it shows promise in preventing complications and improving healing in surgical incisions; therefore, it is necessary to expand this field of knowledge to establish intervention protocols if the results are favorable.

Other topical applications used 1,4DAB or putrescine [26] in the treatment of breast reduction scars; these generate their action by inhibiting transglutaminase and apoptosis of fibroblasts [49]. In the selected study, improvements were found at 6 and 12 weeks in the firmness and characteristics of the scar, which coincides with the claims of a previous review [50].

Growth factors were also used in one of the studies [42], obtaining good results in topical application. This is consistent with systematic reviews and meta-analyses conducted with growth factors applied to other etiologies [51,52]; however, there is little rigorous research on its application to surgical incision healing.

In contrast, one of the studies applied injectable anti-connective tissue growth factor, obtaining very good results at 12 and 24 weeks [29]. However, this is a preliminary study, so it is not known if it is the optimal dose and if it is a cost-effective treatment. Because of this, no additional research has been found with this compound in the literature consulted.

Another intervention that stands out in the selected studies was the injection of botulinum toxin, having been analyzed in four of the studies. This substance has a proven effectiveness in the treatment of other therapeutic conditions such as spasticity or neuropathic pain [53,54,55]. However, it is still under constant review for its application in the healing process, especially in a preventive way. Its action inhibits the production of fibroblasts, and consequently collagen, being able to stop the appearance of scarring complications such as hypertrophy or keloids [56,57].

In the articles included in this review, it was applied to scars of surgical etiology in the first 10 days after the intervention, with a concentration of 5 U in all cases [22,27,35], except in the dose comparison of Chen et al. [24], which found better results in appearance when using 8 U. This dosage and the results obtained are consistent with the conclusions of the limited evidence available in this regard [57,58]. However, one of the investigations [35] also found that the effect had significant differences between patients with a history of hypertrophic scars, which was not found in the previous literature consulted for this comparison.

On the other hand, some studies analyzed the effect of the skin closure method on the surgical scar. One study compared the Donati technique and the subcuticular suture technique, with better short-term assessment by patients towards the subcuticular technique, but no significant differences were obtained for observers [39]. These results are consistent with a previous systematic review, according to which subcuticular sutures may be preferable to interrupted sutures; however, the available research still has many limitations for establishing particular recommendations [59,60].

Three of the studies conducted comparisons between skin closure with suture and with tissue adhesive [25,31,33]. Significant improvements with tissue adhesive were found in healing time, postoperative pain levels, and patient satisfaction [31], but none of them found significant differences in scar characteristics. This contrasts with the literature found, which claims the improvement of scarring appearance with tissue adhesive [61].

Chung et al. [25] also performed a combination of the two methods of skin closure with non-ablative laser treatment, finding that the best results were offered by the combination of suture and laser treatment, and that the groups treated with non-ablative laser showed greater satisfaction. Other research included in this review [30,32,36,37] used ablative and non-ablative laser treatments. These results agree with a previous systematic review [62], according to which, the treatment of surgical scars by PDL, radiofrequency, and ultrasound can improve the texture thickness and appearance and relieve contractures. However, the literature consulted does not value the long-term results, as well as the combination with other techniques, and the samples were small, so it is necessary to expand the research to establish more specific treatment protocols.

This review aimed to understand the techniques that are currently being investigated to improve surgical healing. The time frame selected was reasonable to show the trend of current research in this area. However, and because of this, limitations related to the youth of certain experimental treatments were found, such as the lack of standardization of techniques, small samples, and unicentric designs, as well as limited high-quality evidence in some therapeutic methods. Consequently, studies that support these conclusions were located, but nevertheless did not reach the cut-off point established in the methodological quality review.

Other techniques, on the other hand, were supported by previous scientific evidence as can be seen in the discussion, although they are still present in current research work due to the need for an increase in evidence with methodological quality. Another limitation of the study is due to the limitation of the search to scars of surgical etiology. Although in other situations such as burns there is a greater availability of scientific evidence and some techniques may be common, the scarring characteristics cannot always be compared, and, therefore, conclusions cannot be clearly drawn.

Despite all these limitations, it was observed that the improvement of post-surgical healing is a field of high activity, with promising techniques. Although these techniques must be expanded and replicated in order to establish reliable protocols and recommendations, they can generate an improvement in health care and in the quality of life of patients.

## 5. Conclusions

Improving the quality of surgical scars remains an area of emerging research. In this systematic review, we found numerous current approaches that sought to minimize the signs and symptoms of surgical scarring. Topical application of silicones, enriched creams, tension relief devices, negative pressure bandaging, intradermal applications, skin closure techniques, and laser treatments were reported.

Silicone gels with herbal extracts such as onion should be studied to ensure good results without causing adverse reactions. Scar tension relief devices show benefits in the healing process and appearance. The incisional negative pressure wound therapy, putrescine, growth factors, and anti-connective tissue growth factor show very encouraging results, although the evidence with surgical scars is still very small, and in some cases practically non-existent.

The injection of botulinum toxin generates good results in surgical scars at doses of 5 U, especially in people with a history of scar hypertrophy, and generates better results in scar width with higher doses. Regarding skin closure, patients value the subcuticular suture technique more, although in the long term it does not generate additional benefits. Similarly, tissue adhesive helps with healing time and pain levels, but offers no difference over time compared to suturing. The combination of subcuticular suture and non-ablative laser treatment improves recovery and scarring characteristics and generates patient satisfaction, as well as the use of isolated ablative and non-ablative laser treatments.

However, despite the improvements demonstrated by some approaches, many of the techniques used have not yet consolidated their parameters, times, doses, and specific action protocols. In addition, most existing research is of low quality, with small samples and single-center designs, which may detract from the external validity of the available evidence. It is necessary to generate extensive and quality research on these approaches and establish comparisons that help to converge and determine which technique is the most appropriate in each surgical intervention, so that the impact of scars on the lives of patients can be reduced.

## Figures and Tables

**Figure 1 healthcare-11-02213-f001:**
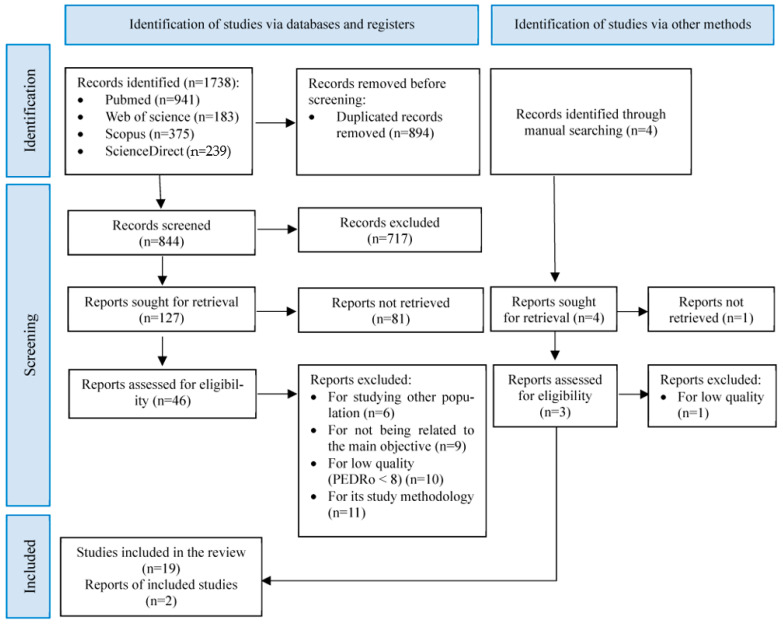
PRISMA flow diagram of search results and included studies.

**Table 1 healthcare-11-02213-t001:** PIO format: keywords.

Population	Surgical Procedures Population
Intervention	Recent approaches of cicatrization process
Outcomes	Quality of scars
Research question	What are the current approaches to improve the quality of scars in surgical procedures population?

**Table 2 healthcare-11-02213-t002:** Search strategy used, adapted to each of the databases.

Database	Search Strategy
PubMed	((“cicatrix” [MeSH Terms] OR “cicatrix” [All Fields] OR “cicatrization” [All Fields] OR “cicatrize” [All Fields] OR “cicatrized” [All Fields] OR “cicatrizing” [All Fields]) AND “Cicatrix, Hypertrophic” [Mesh] AND (prevention) AND (treatment) AND (surgery [Title/Abstract]) OR (surgical scar [Title/Abstract]) NOT (burn* [Title/Abstract]))
Web of Science	((((AB=(cicatrization)) OR AB=(“hypertrophic cicatrix” OR “surgical scar”)) AND AB=(treatment OR therapy)) AND AB=(surgery)) NOT ALL=(conjunctivitis OR burn)
Scopus	TITLE-ABS-KEY (cicatrization) AND TITLE-ABS-KEY (treatment OR therapy) AND TITLE-ABS-KEY (surgery) OR TITLE-ABS-KEY (surgical AND scar)
ScienceDirect	(cicatrization) OR (hypertrophic cicatrix AND prevention) AND (treatment) AND (“surgery” OR “surgical scar” OR) NOT (conjunctivitis OR burn)

**Table 3 healthcare-11-02213-t003:** Results of the PEDro quality assessment of the studies.

	Q1	Q2	Q3	Q4	Q5	Q6	Q7	Q8	Q9	Q10	Q11	Score
Abedini et al., 2020 [22]	Y	Y	Y	Y	Y	Y	Y	Y	Y	Y	Y	11
Chen Z et al., 2022 [23]	Y	Y	Y	Y	N	N	N	Y	Y	Y	Y	8
Chen et al., 2021 [24]	Y	Y	Y	N	Y	Y	Y	Y	Y	Y	N	9
Chung et al., 2021 [25]	Y	Y	Y	Y	N	Y	Y	Y	Y	Y	Y	10
Dolynchuk et al., 2020 [26]	Y	Y	Y	N	Y	Y	Y	N	Y	Y	Y	9
Huang et al., 2019 [27]	Y	Y	Y	Y	Y	Y	Y	N	Y	Y	Y	10
Ilori et al., 2022 [28]	Y	Y	Y	Y	N	N	N	Y	Y	Y	Y	8
Jensen et al., 2018 [29]	Y	Y	Y	Y	Y	Y	Y	Y	Y	Y	Y	11
Karmisholt et al., 2018 [30]	Y	Y	Y	Y	N	N	Y	Y	Y	Y	Y	9
Kong et al., 2020 [31]	Y	Y	Y	Y	N	Y	N	Y	Y	Y	Y	9
Lin et al., 2023 [32]	Y	Y	N	Y	N	N	Y	Y	Y	Y	Y	8
Musham et al., 2023 [33]	Y	Y	Y	Y	N	N	Y	Y	Y	Y	Y	9
Pangkanon et al., 2021 [34]	Y	Y	Y	Y	N	N	Y	Y	Y	Y	Y	9
Phillips et al., 2018 [35]	Y	Y	Y	N	Y	N	Y	N	Y	Y	Y	8
Ramos et al., 2019 [36]	Y	N	Y	Y	Y	N	Y	Y	Y	Y	Y	9
Safra et al., 2019 [37]	Y	N	Y	Y	N	N	Y	Y	Y	Y	Y	8
Surakunprapha et al., 2020 [38]	Y	Y	Y	Y	Y	N	Y	Y	Y	N	Y	9
Suwannaphisit et al., 2021 [39]	Y	Y	Y	N	Y	N	Y	Y	Y	Y	Y	9
Timmermans et al., 2022 [40]	Y	Y	Y	Y	N	N	Y	Y	Y	Y	Y	9
Zhang et al., 2021 [41]	Y	Y	Y	Y	N	N	Y	Y	Y	Y	N	8
Zoumalan et al., 2019 [42]	Y	Y	Y	N	Y	N	Y	N	Y	Y	Y	8

Q1: Eligibility criteria were specified, Q2: Subjects were randomly allocated to groups, Q3: Allocation was concealed, Q4: The groups were similar at baseline regarding the most important prognosis indicators, Q5: There was blinding of all subjects, Q6: There was blinding of all therapists who administered the therapy, Q7: There was blinding of all assessors who measured at least one key outcome, Q8: Measures of at least one key outcome were obtained from more than 85% of the subjects initially allocated to groups, Q9: All subjects for whom outcome measures were available received the treatment or control condition as allocated or, where this was not the case, data for at least one key were analyzed by “intention to treat”, Q10: The results of between-group statistical comparisons are reported for at least one key outcome, Q11: The study provides both point measures and measures of variability for at least one key outcome.

**Table 4 healthcare-11-02213-t004:** Characteristics of the studies included in the systematic review.

Study/Author	Typology/Main Objective	Participants	Interventions	Evaluation	Main Findings	PEDro
Abedini et al., 2020 [22]	Randomized, split-scar, double- blinded, prospective, controlled trial. To investigate the role of botulinum toxin on the prevention of scar formation in comparison with control in cosmetic plastic surgeries including mammoplasty and abdominoplasty.	n = 19 Patients of mammoplasty and abdominoplasty. 26–54 years old. Sex (f/m): 19/0	Botulinum toxin (EG) or saline solution (CG) on each side of scar.	mSBSES Assessments at 3 and 6 months	Significant improvements in EG compared to CG at 3 and 6 months (*p* < 0.001 each). Also, in the subset analysis, there were significant differences between groups in width, height, color, and scar visibility at months 3 and 6. In EG, scores of mSBSES, height, visibility, and redness (*p* < 0.001; *p* = 0.002; *p* = 0.002; *p* = 0.008) increased significantly from month 3 to 6, but the scar width did not change significantly (*p* = 0.051). In CG scores of mSBSES, height, and redness (*p* = 0.0015; *p* = 0.038; *p* = 0.019) improved significantly over time, but scar width and visibility (*p* = 0.34; *p* = 0.24) did not change significantly.	11
Chen et al., 2022 [23]	Randomized, controlled, prospective, rated blinded trial. To determinate efficacy and safety of tissue adhesive zippers in post-surgical scar prevention among patients undergoing surgical excision of the face.	n = 53 Patients with surgical excision on the face. <14 years old. Sex (f/m): 30/23	EG participants used a tissue adhesive zipper for three months. CG no intervention.	Scar Width. POSAS Assessments at 1, 3, 6, 12 months.	In EG group the scar width was significantly smaller than CG (*p* = 0.0025) at 12 months. EG and CG differed significantly in POSAS scores for Scar irregularity (*p* = 0.0145). No differences between groups in the other scores and observer score.	8
Chen et al., 2021 [24]	Randomized, prospective, double-blind, split-scar trial. To investigate the effect of different doses of botulinum toxin administered early after surgery on scar improvement through a split-scar experiment.	n = 22 Tumor resection (tumor did not invade the muscle) 18–52 years old. Sex (f/m): 9/11	High and low doses of botulinum toxin into each half of the surgical wound closure.	mSBSES. VAS. Assessment at 6 months	The high-dose sides had significantly better mSBSES score compared with low-dose in terms of width (*p* < 0.01), incision visibility line (*p* < 0.01). No significant differences between groups in height and color. High-dose sides had significantly higher VAS scores than low-dose sides (*p* < 0.01).	9
Chung et al. 2021 [25]	Randomized, blinded, prospective trial. To compare the scar quality when different protocols were applied, and eventually aim to find the optimal scar management protocol	N = 126 Patients undergoing thyroidectomy. >18 years old Sex (f/m): 105/21	Tissue adhesive (group A), or subcuticular suturing and early NAFL (group B), or skin closure with tissue adhesive and early NAFL (group C)	POSAS. Assessment at 6 months.	No significant differences between groups at baseline. At 6 months, B group showed a narrower width scar, with no differences between A and C (*p* > 0.017). According to the patients, groups B and C showed significantly higher satisfaction in all sub-scales than A except for the pigmentation. According to physicians, B showed better thickness, relief, pliability, surface area, and overall cosmesis (*p* < 0.017) with no differences between A and C (*p* > 0.017).	10
Dolynchuk et al., 2020 [26]	Randomized, prospective, double-blind trial. To analyze the biochemical and clinical effects of 1,4-Diaminobutane (DAB)on prevention of human hypertrophic scars.	Total n = 78 patients of breast reduction. Biochemical evaluation n = 30 Clinical evaluation n = 48 5–53 years old. Sex (f/m): 50/10	Topical 1,4-DAB or control treatment on each side (EG or CG respectively).	Biochemical: Analytical assessment (biopsy) At 2 months Clinical: Durometer test POSAS At 6 and 12 weeks	The biopsies registered more 1,4DAB in treated scars than control group (*p* < 0.05). Durometer test was significantly better (*p* < 0.05) in EG. POSAS score was significantly better in EG than CG (*p* < 0.05)	9
Huang et al., 2019 [27]	Randomized, prospective, double blind, split-face trial. To investigate the safety and efficacy of early botulinum toxin A injection in preventing hypertrophic scarring in the medial cantal area after epicanthoplasty.	n = 43 Patients of epicanthoplasty. 18–45 years old. Sex (f/m): 43/0	Botulinum toxin A (EG) or saline treatment (placebo side) into each side of surgery.	VSS VAS. Patient Satisfaction Assessment al 1, 3, 6 months	13 patients were lost to follow-up. The botulinum toxin A side had significantly better scores at VSS 1 month (*p* = 0.034), 3 months (*p* < 0.001), and 6 months (*p* < 0.001) after administration. The same was found at VAS (*p* = 0.017; *p* < 0.001; *p* = 0.032 respectively). Patient satisfaction was better with botulinum toxin A (*p* < 0.001)	10
Ilori et al., 2022 [28]	Randomized, controlled, prospective trial. To determinate the efficacy of microporous tape in the prevention of abnormal post-surgical scars.	n = 72 patients with 92 scars EG n = 36 CG n = 36 Benign tumors excision, open reduction fixation of fractures, osteotomies, arthroplasties. 15–65 years old. Sex (f/m): 25/38	Microporous tape directly over the scar for 6 months (EG) or standard care.	POSAS Scar types (normal, hypertrophic, or atrophic). Assessments at 6 weeks, 3 months, and 6 months.	No significant differences between groups at the baseline. At 6 months, the scar height and width were significantly better in EG than CG (*p* < 0.0001 each), and scar types were significantly better in EG than CG (*p* = < 0.0001).	8
Jensen et al., 2018 [29]	Randomized, double-blinded, within-subject, placebo-controlled, prospective trial. To determinate the effect of anti-CTFG (EXC001) on the severity of surgical scars.	n = 23 Mammoplasty. Bilateral, symmetric hypertrophic scars of the breast. 28–55 years old. Sex (f/m): 23/0	EXC001 (EG) or placebo (CG) injected intradermally at post-surgery weeks 2, 5, 8, and 11	POSAS Assessment at 12 and 24 weeks.	At 24 weeks, EG reduced scar severity significantly compared to CG, by physician (vascularity *p* < 0.001; pigmentation *p* < 0.001; thickness *p* = 0.001; relief *p* < 0.001; Pliability *p* = 0.005; surface area *p* < 0.001; overall opinion *p* < 0.001) and subjects (pigmentation *p* = 0.01; stiffness *p* = 0.003; thickness *p* = 0.005; surface area *p* = 0.032 and overall opinion *p* = 0.003)	11
Karmisholt et al., 2018 [30]	Randomized split-wound, prospective trial. To assess scar formation clinically after three nonablative fractional laser (NAFL) exposures, targeting the inflammation, proliferation, and remodeling wound healing phases in patients vs. untreated controls.	n = 32 Patients undergoing surgical excision. >18 years old Sex (f/m): 15/17	1540-nm NAFL (3 exposures: before surgery, at suture removal and 6 weeks after surgery) (EG) or no laser treatment on each side of the scar (CG).	POSAS VSS Assessment at 3 months	30 patients completed follow-up. At 3 months, EG improved scores compared with the control in POSAS, vascularity, relief, pliability, surface area, and overall opinion (*p* < 0.001; *p* = 0.005; *p* = 0.023; *p* = 0.037; *p* = 0.016; *p* = 0.003), but pigmentation and thickness did not (*p* = 0.13 each). The greater improvements were located in thorax area (*p* < 0.001). Patients older and younger than 50 years answered similarly (*p* = 0.015; *p* = 0.008)	9
Kong et al., 2020 [31]	Randomized, prospective, controlled trial. To present the experience of adopting tissue adhesive as adjunct to standard wound closure in total hip arthroplasty and evaluate its role and cost performance.	n = 30 Patients with bilateral total hip arthroplasty. 18–60 years old. Sex (f/m): 13/17	Standard wound closure (GC) or additional tissue adhesive (EG).	PSAS HWES VSS Assessment at 1 month	PSAS showed that, from the view of patients, hips with tissue adhesive were significantly better than sutured hips (*p* = 0.004). Most patients preferred the tissue adhesive. From view of evaluators, there were not significant differences between groups in HWES or VSS.	9
Lin et al., 2023 [32]	Randomized, prospective, single blinded, split-scar trial. To compare surgical scars treated with fractional carbon dioxide (CO_2_) laser performed on Day 0 and Day 14.	n = 30 Patients of skin cancer excision on limbs. 34–82 years old Sex (f/m): 13/17	2 passes of CO_2_ laser before the cutaneous suture (day 0) or when sutures were removed (day 14) on each side.	MSS Analytical assessment (biopsy) Assessment at 6 months	26 subjects completed follow-up. There were no differences between groups for patients (*p* = 0.058) or physicians (*p* = 0.028). Fractal dimensions and lacunarity were similar (*p* = 0.80; *p* = 0.44).	8
Musham et al., 2023 [33]	Randomized, prospective, controlled, single-blind trial. To compare the skin closure time, postoperative pain, and the scar outcome between tissue adhesive and sub-cuticular sutures in thyroid surgery.	n = 124 Patients of thyroidectomy. 30–55 years old Sex (f/m): 96/28	Subcuticular suture (CG) or tissue adhesive (EG)	VAS MSS Assessment at post operative, 1, and 3 months.	VAS showed better post operative results in EG (*p* < 0.01). However, there were not significant differences between groups in MSS at 1 and 3 months.	9
Pangkanon et al., 2021 [34]	Randomized, assessor-blind, prospective controlled trial. To compare the efficacy of silicone gel containing onion extract and aloe vera (SGOA) to silicone gel sheets (SGS) to prevent postoperative hypertrophic scars and keloids.	n = 40 Patients underwent surgery. 18–60 years old Sex (f/m): 36/4	SGOA twice a day or SGS wound 24 h/day.	POSAS Mexameter Cutometer Assessments at 1, 2, and 3 months.	No differences between groups in number of hypertrophic scars (*p* = 0.465), melanin (*p* = 0.571), erythema (*p* = 0.863), and POSAS by physicians and patients. SGOA group had significantly greater pliability (*p* = 0.009).	9
Phillips et al., 2018 [35]	Randomized, double blind, prospective controlled trial. To assess the effects of botulinum toxin type A on scar formation after thyroid surgery	n = 40 Patients of total thyroidectomy, hemithyroidectomy, or parathyroidectomy. Sex (f/m): 36/4	Botulinum toxin A (EG) or saline solution (CG) on each half of the scar.	POSAS VSS Preoperative evaluation. Assessments at 1, 6, and 12 months.	23 patients completed the last follow-up. There were not significant differences between groups at 1, 6, and 12 months. However, at 6 months, better results were found in EG compared to CG in patients with poor cicatrization history in POSAS total score (*p* = 0.012), overall (*p* = 0.010), and VSS (*p* = 0.021)	8
Ramos et al., 2019 [36]	Non-randomized, double-blinded split-scar prospective trial. To evaluate the influence of photobiomodulation on the post abdominoplasty healing process.	n = 17 Patients of abdominoplasty 18–55 years old Sex (f/m): 17/0	Right side of scars: 10 sessions of photobiomodulation (experimental side). Left side of scars: any treatment (used as control).	POSAS VSS Assessments at 1, 6, and 12 months.	The treated side of scars was significantly better after 1 and 6 months on VSS (*p* = 0.0065). The scores of POSAS were better on treated side for observers (*p* = 0.0034) and patients (*p* = 0.0047).	9
Safra et al., 2019 [37]	Randomized, controlled, single blinded, split-scar, prospective trial. To study the safety and efficacy of a combination of pulsed dye laser (PDL) and fractional ablative CO_2_ laser (FACL) for attenuation of post-lumpectomy scarring.	n = 18 Patients of lumpectomy. >18 years old Sex (f/m): 18/0	Treated side received 3 sessions at 1-month intervals of PDL and FACL 6 weeks after suture removal. The other half of scars did not receive any treatment	POSAS Assessment at 6 months.	The improvements in scar parameters were significantly greater in treated side (overall, pigmentation, and relief *p* < 0.001; vascularity and pliability *p* = 0.001; thickness *p* = 0.002).	8
Surakunprapha et al., 2020 [38]	Randomized, controlled, double-blind, prospective trial. To determine whether adding herbal extracts to the gel would augment the healing effect.	n = 46 Patients of sternotomy 32–61 years old. Sex (f/m): 19/27	EG: topical silicone gel plus herbal extract gel (Allium cepa, Centella Asiatica, Aloe vera and Paper Mulberry). CG: topical silicone gel	POSAS Assessments at 6 months	At 6 months, EG had significantly greater scores than baseline: vascularity (*p* = 0.013), pigmentation (*p* = 0.000), overall (*p* = 0.018). CG also had improvements in pigmentation (*p* = 0.000) and vascularity (*p* = 0.046).	9
Suwannaphisit et al., 2021 [39]	Randomized, prospective controlled trial. To compare the Donati suture technique and running subcuticular technique in terms of surgical scar, pain, and functional outcome.	n = 142 Patients of open carpal tunel release. 48–70 years old. Sex (f/m): 120/18	Donati or running subcuticular technique.	POSAS Pain (verbal numerical rating score) Assessments at 2, 6, and 12 weeks	At 2 weeks, POSAS showed that subcuticular running technique had lower scores than Donati by the patients (*p* < 0.05) but not by physicians (*p* = 0.15). At 6 and 12 weeks, there was no difference between groups in any parameter.	9
Timmermans et al., 2022 [40]	Randomized, prospective, within- subject controlled trial. To establish if incisional negative pressure wound results in improved scar outcomes in comparison to the standard of care.	n = 85 Transgender men undergoing gender-affirming mastectomies. 18–63 years old Sex (f/m): 0/85	Incisional negative pressure wound (EG) or standard care (CG) on each side.	Cutometer. POSAS Assessments at 1, 3, and 12 months.	80 patients completed follow-up. At 12 months, there were not significant differences between groups for Cutometer subdomains (*p* > 0.05). Significant improvements were found in EG compared to CG at 3 months in vascularity, POSAS total score, and overall cosmesis (*p* = 0.022; *p* = 0.003; *p* = 0.004), but these differences were not found at 12 months. By patients, at 1 and 3 months, thickness was better in EG (*p* = 0.027; *p* = 0.042), and at 12 months color, pliability, thickness, total score, and overall opinion were better in EG (*p* = 0.003; *p* < 0.001; *p* = 0.003; *p* = 0.039; *p* = 0.008).	9
Zhang et al., 2021 [41]	Randomized, single blind, prospective controlled trial. To determine whether the application of a tension offloading device preoperatively would result in superior attenuation of scar genesis in comparison to traditional methods.	n = 12 Patients with a history of hypertrophic scar formation, who underwent surgical excision of benign cutaneous lesions located over buttocks and truncal region. 11–33 years old Sex (f/m): 8/4	Application of device before (2 weeks) and after surgery (for 3 months, starting from the suture removal) (EG1); application after surgery (for 3 months) (EG2); or no tension offloading (CG)	Skin elasticity coefficient Assessment Before surgery POSAS Assessment at 6 months	The median skin elasticity coefficient was 27.5% in the pre-op group in comparison to 15% in both the post-op group and the control group (*p* = 0.0286). EG1 and EG2 did not show significant differences. There were differences compared to CG in terms of width and color. Overall Score was significantly better between 3 groups by patients but only between offloaded groups and CG by physicians.	8
Zoumalan et al., 2019 [42]	Randomized, double-blinded, multicenter, prospective trial. To compare the efficacy and safety of a scar cream consisting on highly selective human growth factors (SKN2017B) and hyaluronic acid within a silicone matrix.	n = 45 (49 bilateral and 12 unilateral scars) Surgical patients with unilateral or bilateral scars on their face. >18 years old. Sex (f/m): 43/2	SKN2017B or silicone cream twice for 3 months. Unilateral scars were randomly assigned in split-scar and bilateral scars were assigned by sides.	VSS Assessment at 12 weeks. (Independent Assessment)	Investigators rated 74% improvements in SKN2017B group and 54% in silicone group (*p* < 0.0001), patients rated 85% and 51% improvements respectively (*p* < 0.001). Independent reviewers rated 87% and 1% (*p* < 0.0001).	8

POSAS (Patient Observer Scar Assessment Scale Score); mSBSES (Modified Stony Brook Scar Evaluation Scale); VAS (Visual Analogue Scale); VSS (Vancouver Scar Scale); MSS (Modified Manchester Scar Scale); EG (Experimental Group); CG (Control Group).

## Data Availability

All the data are in the manuscript.

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
