# Peer review of "Treatment and Improvement of Healing after Surgical Intervention"

_healthcare, 2023, doi:10.3390/healthcare11152213_

Round 1

Reviewer 1 Report

Dear Authors, 

The subject of the manuscript is very interesting.

I think that in ,,material and methods,, should clarify that the authors aimed to evaluate in each article of the study.

I think in results should be better organized and clearer regarding the anatomical region operated with their particularities, the type of sutures, the suture materials and the substances or methods of the treatment used for the scars.

I encourage the authors to reevaluate the editing of the manuscript in accordance with the ,,instructions for authors,,.

Thank you!

Author Response

Dear reviewer,

Thank you very much for inviting us to resubmit a revised version of our manuscript entitled: “Treatment and improvement of healing after surgical intervention. Systematic review”. We have checked our manuscript according to your comments, the quality of English, and the Journal requirements. (We have also responded below to some comments from reviewers point by point).

We would be very grateful if you could consider our manuscript to be published in your journal.

We have included a revised manuscript file v.2.

Yours sincerely,

Dr Rocío Pardo-Hernández

I think that in ,,material and methods,, should clarify that the authors aimed to evaluate in each article of the study.

Response: thank you so much for pointing it out. It was modified in the text, including the evaluation criteria for each study, and it was also added to the explanation of table 3.

I think in results should be better organized and clearer regarding the anatomical region operated with their particularities, the type of sutures, the suture materials and the substances or methods of the treatment used for the scars.

Response: thank you so much for pointing it out. To clarify the information summarized in the results we have added information about the suture materials and the timing of treatments. However, given the wide variability between methods, times and materials, the comparison between types of intervention has had to be maintained. We hope that in this way some details of the studies will be better appreciated.

I encourage the authors to reevaluate the editing of the manuscript in accordance with the ,,instructions for authors,,.

Response: Thank you for pointing it out. We have checked the instructions for authors and correct the manuscript in accordance with that. 

Reviewer 2 Report

The submitted manuscript “Treatment and improvement of healing after surgical intervention. Systematic review” describes well the intention from the authors to identify recent advances in treatment modalities for improving scar outcome following surgical intervention. Nevertheless, there are a number of issues in the current review I believe should be addressed prior to resubmission and as such cannot recommend publication at this time.

The introduction should articulate more clearly what the current treatment modalities (49-55) consist of, if these are in response to the known risk factors (listed on 44-46) and how these differ from the ‘new research’ captured by the current review.

While the materials describe much of the method for the systematic review, they would benefit from an additional table articulating the quality assessment questions (identified in table 3 simply as Q1 – Q11) which should line up and expand on lines 72-79 and 88-94 which currently lists article selection criteria (4 parameters) and data extraction aspects (8 aspects) but not how the scientific quality was evaluated.

The presentation of the results of the review would benefit from restructuring to summarise the different approaches into those which are carried out prior to or at the time of the original surgical procedure (eg suture techniques, off-loading devices), treatments which occur during scar formation (eg topical applications or injections into scar) and those which are more revisionary techniques once a scar has formed. In particular, the term ‘differences’ should always be qualified to articulate if the new treatment is better than the control as some statistical differences between groups may actually not be improvements. This should be in the results and discussion as well as table 4 which should also mention that (EG) is not an abbreviation for Botulinum toxin and (CG) is not an abbreviation for the solution as the first entry indicates, rather I assume it stands for Experimental group (EG) and control group (CG) as it is used through the table.

The limitations of the current review should also be discussed, particularly around the exclusion of burn etiology scar treatment studies and some discussion about if learnings from this research area could be translated across to scars arising from surgical scars. Additionally, justification for the use of the term cicatrization’ rather than the more simple ‘scar formation’ should be included. Moreover, the authors have limited the review period to articles published in the last 5 years, however, in the discussion there is comparison to studies carried out prior to this and incorporated in other meta reviews (eg as referenced in the discussion ref 58). Additionally the discussion compares results from the studies captured in this current systematic review to those in other studies (eg references 53-57) which have publication dats of 2019 – 2022 and as such it is unclear why they were excluded from analysis. This should be commented on.

Overall, the discussion is where the largest improvements can be made, and where the authors should give some indications as to what are the differences between trials in terms of treatment regime and the final conclusions should be around areas of greatest potential. 

In general, there are a many language issues that should be corrected throughout the manuscript. As an example, in just the first few paragraphs there is the incorrect use of the word fabric to describe the dermis post healing on line 36, “it has been tried to discover the mystery of the biological mechanisms” line 43 and “it is essential to know and meticulous planning of the incisions” line 52. Moreover, there is one line that is not in English at all (line 217-218) and the use of ‘y’ rather than ‘and’ on line 66.

Author Response

Dear reviewer,

Thank you very much for inviting us to resubmit a revised version of our manuscript entitled: “Treatment and improvement of healing after surgical intervention. Systematic review”. We have checked our manuscript according to your comments, the quality of English, and the Journal requirements. (We have also responded below to some comments from reviewers point by point).

We would be very grateful if you could consider our manuscript to be published in your journal.

We have included a revised manuscript file v.2.

Yours sincerely,

Dr Rocío Pardo-Hernández

The introduction should articulate more clearly what the current treatment modalities (49-55) consist of, if these are in response to the known risk factors (listed on 44-46) and how these differ from the ‘new research’ captured by the current review.

Response: thank you so much for pointing it out. It was revised according to the recommendations of the reviewers.

While the materials describe much of the method for the systematic review, they would benefit from an additional table articulating the quality assessment questions (identified in table 3 simply as Q1 – Q11) which should line up and expand on lines 72-79 and 88-94 which currently lists article selection criteria (4 parameters) and data extraction aspects (8 aspects) but not how the scientific quality was evaluated.

Response: thank you so much for pointing it out. It was modified in the text, including the evaluation criteria for each study, and it was also added to the explanation on footer of table 3.

The presentation of the results of the review would benefit from restructuring to summarise the different approaches into those which are carried out prior to or at the time of the original surgical procedure (eg suture techniques, off-loading devices), treatments which occur during scar formation (eg topical applications or injections into scar) and those which are more revisionary techniques once a scar has formed. In particular, the term ‘differences’ should always be qualified to articulate if the new treatment is better than the control as some statistical differences between groups may actually not be improvements. This should be in the results and discussion as well as table 4 which should also mention that (EG) is not an abbreviation for Botulinum toxin and (CG) is not an abbreviation for the solution as the first entry indicates, rather I assume it stands for Experimental group (EG) and control group (CG) as it is used through the table.

Response: thank you very much for pointing it out. To clarify the information summarized in the results we have added information about the suture materials and the timing of treatments. However, given the wide variability between methods, times and materials, the comparison between types of intervention has had to be maintained. We hope that in this way some details of the studies will be better appreciated. Capital letters EG and CG are now mentioned on footer of table 4. The term “differences” has been changed to clarify the effect of treatments.

The limitations of the current review should also be discussed, particularly around the exclusion of burn etiology scar treatment studies and some discussion about if learnings from this research area could be translated across to scars arising from surgical scars. Additionally, justification for the use of the term cicatrization’ rather than the more simple ‘scar formation’ should be included. Moreover, the authors have limited the review period to articles published in the last 5 years, however, in the discussion there is comparison to studies carried out prior to this and incorporated in other meta reviews (eg as referenced in the discussion ref 58). Additionally the discussion compares results from the studies captured in this current systematic review to those in other studies (eg references 53-57) which have publication dats of 2019 – 2022 and as such it is unclear why they were excluded from analysis. This should be commented on.

Overall, the discussion is where the largest improvements can be made, and where the authors should give some indications as to what are the differences between trials in terms of treatment regime and the final conclusions should be around areas of greatest potential. 

Response: Thank you for pointing it out. The discussion has been expanded with the necessary limitations and clarifications.

Comments on the Quality of English Language

In general, there are a many language issues that should be corrected throughout the manuscript. As an example, in just the first few paragraphs there is the incorrect use of the word fabric to describe the dermis post healing on line 36, “it has been tried to discover the mystery of the biological mechanisms” line 43 and “it is essential to know and meticulous planning of the incisions” line 52. Moreover, there is one line that is not in English at all (line 217-218) and the use of ‘y’ rather than ‘and’ on line 66.

Response: thank you so much for pointing it out. We have corrected the translation errors.

Reviewer 3 Report

This review is interesting and well written and introduces a topic of crucial importance for plastic surgeons, namely being able to guarantee the patient a good scar after surgery.

The authors should improve some points of their manuscript:

- introduction. This is poor in information especially in the pathological healing process and what are the biological and molecular phases that lead to the hypertrophic scar.

- the authors should cite an important work that was published by De Francesco F et al on Aesthetic Plastic Surgery where the authors use hyaluronic acid as a modulator of hypertrophic scars. Please cite the article and comment on it.

Author Response

Dear reviewer,

Thank you very much for inviting us to resubmit a revised version of our manuscript entitled: “Treatment and improvement of healing after surgical intervention. Systematic review”. We have checked our manuscript according to your comments, the quality of English, and the Journal requirements. (We have also responded below to some comments from reviewers point by point).

We would be very grateful if you could consider our manuscript to be published in your journal.

We have included a revised manuscript file v.2.

Yours sincerely,

Dr Rocío Pardo-Hernández

- introduction. This is poor in information especially in the pathological healing process and what are the biological and molecular phases that lead to the hypertrophic scar.

Response: thank you so much for pointing it out. It was revised according to the recommendations of the reviewers.

- the authors should cite an important work that was published by De Francesco F et al on Aesthetic Plastic Surgery where the authors use hyaluronic acid as a modulator of hypertrophic scars. Please cite the article and comment on it.

Response: thank you very much for pointing it out. This reference was included and mentioned in the current approaches of abnormal scars.

Round 2

Reviewer 2 Report

Thank you for addressing the comments from my initial review. This has greatly improved the quality of the manuscript. The only remaining issue is in the initial paragraph of the introduction which appears to indicate the primary outcome of the remodelling phase to be reepithelisation (which actually often occurs in the proliferation phase, though continued stratification can occur in the remodelling) and should be corrected. The remodelling phase generally is considered to be the phase during which the dermis (rather than the epidermis) is remodelled and scar formation occurs.

Author Response

Dear reviewer,

Thank you very much for inviting us to resubmit a revised version of our manuscript entitled: “Treatment and improvement of healing after surgical intervention. Systematic review”. We have checked our manuscript according to your comments. We have also responded below to your comments.

We would be very grateful if you could consider our manuscript to be published in your journal.

We have included a revised manuscript file v.3.

Yours sincerely,

Dr Rocío Pardo-Hernández

Thank you for addressing the comments from my initial review. This has greatly improved the quality of the manuscript. The only remaining issue is in the initial paragraph of the introduction which appears to indicate the primary outcome of the remodelling phase to be reepithelisation (which actually often occurs in the proliferation phase, though continued stratification can occur in the remodelling) and should be corrected. The remodelling phase generally is considered to be the phase during which the dermis (rather than the epidermis) is remodelled and scar formation occurs.

Thank you for pointing out this error. The paragraph has been redrafted to correct it.

Reviewer 3 Report

The authors issues my previous concern. The authors should also include in the table the paper by Riccio M et al on the use of hyaluronic acid

Author Response

Dear reviewer,

Thank you very much for inviting us to resubmit a revised version of our manuscript entitled: “Treatment and improvement of healing after surgical intervention. Systematic review”. We have checked our manuscript. We have also responded below to your comments.

We would be very grateful if you could consider our manuscript to be published in your journal.

We have included a revised manuscript file v.3.

Yours sincerely,

Dr Rocío Pardo-Hernández

The authors issues my previous concern. The authors should also include in the table the paper by Riccio M et al on the use of hyaluronic acid

Response: Thank you so much for your comment. We have included previously that article in the introduction. However, this article was not found during the search process with Boolean terms and operators, nor during manual reverse lookup. As a result, it could not be included in the tables and comments. Including it would alter the process of a systematic review, becoming a literature review, because the process could not be replicated.